# Acute Heart Failure, 90-Day Mortality, and Gravitational Ischemia in the Brain

**DOI:** 10.3390/diagnostics12061473

**Published:** 2022-06-15

**Authors:** J. Howard Jaster, Giulia Ottaviani

**Affiliations:** 1London Corporation, London SW7 1EW, UK; harbert38104@yahoo.com; 2Anatomic Pathology, Lino Rossi Research Center, Department of Biomedical, Surgical and Dental Sciences, Università degli Studi di Milano, 20122 Milan, Italy

**Keywords:** heart failure, gravity, ischemia, brain

## Abstract

During the 90 days following hospitalization for acute heart failure, the ejection fraction and type of discharge medications have been shown in clinical trials to have little effect on mortality. We examined the recent literature addressing brain-related etiologies of sudden death following heart failure. Two mechanisms of sudden unexpected death have been suggested to possibly result from four significant influences on pathophysiology in the brain. The two causes of sudden death are (1) severe cardiac arrhythmia and (2) neurogenic pulmonary edema. They are both mediated through the brainstem autonomic nuclei generally and executed specifically through the dorsal motor nucleus of the vagus nerve. The four significant influences on pathophysiology, all contributing to ischemia in the brainstem autonomic nuclei, are: (1) Hyper-stimulation of neurons in the solitary tract nucleus, increasing their metabolic requirements; (2) Inadequate blood flow at a vascular watershed terminus, perfusing the autonomic nuclei; (3) Additionally decreased blood flow, globally throughout the brain, following vasoconstriction related to relative hyperventilation and decreased pCO_2_ levels; (4) Gravitational ischemia in the brainstem caused by the weight of the cerebral hemispheres sitting above the brainstem. Changes in head tilt release gravitational ischemia in the brain. There is no specific head position relative to gravity that is considered favorable or unfavorable for an extended period of time, longer than 24 h. Even a small degree of head elevation, used in managing pulmonary congestion, may increase gravitational ischemia in the posterior fossa and brainstem. In this paper, we suggest a new and important research avenue for intervening in cardiac arrhythmias and preventing their occurrence through the non-invasive use of head-tilting and other head repositioning maneuvers. This could potentially help many geriatric patients with heart failure, who have decreased mobility in bed, and who tend to stay in one position longer, thereby increasing gravitational ischemia in the brain.

## 1. Introduction

Recently, a multi-center group of Italian investigators [1] evaluated patients with pulmonary and intravascular congestion at admission, as well as repeatedly during hospitalization, for acute decompensated heart failure. Patients with a reduced ejection fraction (*n* = 142) were compared to those with a preserved ejection fraction (*n* = 172) using lung ultrasound and inferior vena cava ultrasound. This prospective study included 314 patients, mostly around age 80. Primary outcomes included death or re-admission to the hospital for heart failure at 90 days. Cogliati et al. [1] concluded that there was no significant difference in the primary outcome between the two groups. They further suggested that “other factors beyond ejection fraction could play a role in congestion/decongestion patterns”. This was consistent with previous reports by Van Aelst et al. [2].

The results of the study of Cogliati et al. [1] echoed the results of another study reported by Spanish investigators just a few months earlier [3]. That study, as well, looked at heart failure patients, but only the subset with preserved ejection fraction upon discharge from the hospital, specifically regarding their discharge medications. Tost et al. [3] compared the patient medication profiles with their primary outcomes of death or re-admission to the hospital for heart failure at 90 days. Specifically, they studied the use of anti-neurohormonal drugs, including beta-blockers, renin–angiotensin–aldosterone system inhibitors, and mineral–corticosteroid receptor antagonists. The patients were grouped according to whether they did (*n* = 2312) or did not (*n* = 993) take any of these medications following discharge. This multi-center retrospective study included 3305 patients, around 83 years old. Tost et al. [3] concluded that there was no significant difference in primary outcomes between the two groups.

The results from both of these studies [1,3] may seem to defy logic. It may seem counter-intuitive to some that heart failure patients with a poor ejection fraction [1] have the same 90-day mortalities as those with preserved ejection fraction or that heart failure patients using medications known to improve hemodynamics and cardio-dynamics [3] do not experience a benefit in 90-day mortality following hospital discharge.

These discrepancies may have been resolved two decades earlier by a group of Italian Neuroscience–Pathophysiology researchers [4], who studied five autopsy cases, comparing the brainstems of patients who died of acute heart failure. Their focus was the solitary tract nuclei, a component of the autonomic system, which receives messages about cardio-vascular and respiratory parameters from all over the body, mostly through the vagus network.

## 2. Discussion

### 2.1. Excitatory Sensory Ischemia in a Vascular Watershed Terminus

De Caro et al. [4] discovered ischemia and infarctions in the brainstem, but they were limited to the solitary tract nuclei and did not involve a typical vascular distribution. They partly attributed this seemingly odd pattern to intense hyper-excitatory sensory neuronal input into the solitary tract nuclei during heart failure, which increased neuronal cellular metabolic requirements there. They also cited the somewhat reduced blood flow to this region, which is anatomically located at the end of a watershed area and may be insufficient when metabolic requirements during hyper-excitation become very increased there [4].

These two factors, taken together, may predispose this area of the brainstem to ischemia, even in the absence of what is typically thought of as ‘occlusive cerebrovascular disease’. Although subsequently cited dozens of times, the concepts outlined by De Caro et al. in [4] have probably been under-utilized in clinical practice, as is possibly demonstrated by the two recent papers [1,3], neither of which mentioned the possible role of the brainstem. The concepts [4] additionally provided reasonable answers [5] to questions previously unresolved regarding arrhythmia-induced sudden unexpected deaths related to small ischemic brain lesions in the medulla, the home of the solitary tract nucleus. Someday they may allow us to intervene non-invasively in medical management to improve outcomes by preventing cardiac arrhythmias generated in the brain.

### 2.2. Carbon Dioxide, Gravity, and Thinking like a Skin Nurse

More recently, two additional contributing bio-physical factors acting on blood flow in the brainstem have been suggested as having a significant clinical role in the setting of heart failure. They seem capable of triggering abnormal discharges from the dorsal motor nucleus of the vagus nerve—the major outflow component of the vagus system, among the brainstem autonomic nuclei. This may initiate severe cardiac arrhythmias.

These two additional factors are vasoconstriction in the brain related to carbon dioxide (CO_2_) levels [6] and gravity in the brain [7,8]. These possibly have several clinical effects, but the one applicable here is sudden unexpected death related to both cardiac arrhythmias and neurogenic pulmonary edema [7,8], and are possibly responsible for significant mortality in patients discharged from the hospital following acute heart failure.

CO_2_ is a potent vasodilator in the brain, and reduced CO_2_ levels typically cause vasoconstriction, thereby contributing to ischemia in brainstem autonomic nuclei [6]. CO_2_ levels become decreased most frequently as a result of mechanical ventilation or respiratory therapy. They may similarly become decreased during the management of sleep apnea, a frequent co-morbidity of heart failure during the use of continuous positive airway pressure (CPAP).

Then, there is gravity. Encased in the skull, the brain is one of the least mobile and least accessible organs in the body. The external surfaces of the brain lie still against the relatively hard inside surfaces of the skull. The meninges and cerebrospinal fluid surrounding the brain may provide some cushioning but do not mitigate the effects of gravity. In contrast, the heart and lungs are continuously in motion, and they are surrounded to a significant degree by soft tissues.

Functional gravity-centered ischemia in the brain is induced by the mass of one brain component acting upon another in Earth’s gravitational field [7,8,9]. With any degree of head tilting, the ‘upper’ portion of the brain (relative to Earth’s center of gravity) is sitting on the ‘lower’ portion as a burden of weight. For most people, their head and body orientations are approximately upright for most of the day and then roughly horizontal for 7 h at night during sleep. Ischemia, which may occur on the lower layers, can be reversed in its earlier stages (Figure 1).

In healthy individuals, the horizontal supine body positioning associated with sleep helps to redistribute both gravitational ischemia and blood flow after a 16 h period of vertical head positioning during the waking hours of the day. Restoration of blood flow by reopening capillary vascular beds follows repositioning (unloading of ischemic regions) of the brain relative to gravity by head tilting, which is significant through the 24 h sleep/wake cycle [7,8,9].

Gravitational ischemia in the brain may potentially be largely preventable by frequently changing the head tilt, just as ischemic skin breakdown, bed sores, and decubitus ulcers are currently prevented by frequent changes in general body positioning, focused on the effects of gravity. Skin nurses currently implement these changes around the clock in hospitals around the world.

### 2.3. Can the Effects of Gravity Be Seen on Brain Imaging?

Recently, Welsh investigators [10] reported that the sagging of the brain under the influence of gravity, called ‘positional brain shift’, is important in relation to the specific geometric coordinates established by magnetic resonance imaging (MRI) for the purpose of orienting stereotactic surgery. If the orienting coordinates are established by an MRI performed in the supine position, the brain will shift slightly away from those coordinates if the patient is placed into a different position for surgery. This was discovered by examining 11 healthy adults, who were moved from a supine position to a prone position, and then back again into a supine position [10].

Zappalà et al. [10] discovered that a small, unsymmetrical brain shifting due to differences in head tilting could result in a significant difference between the intended and the actual location of surgical intervention. Additionally, the strain analysis component of their examination revealed regional variations in ‘compressibility’ within the brain. When horizontally rotating from the prone position back to supine, the anterior regions of the brain showed expansion (generated by changes in both volume and shape), and the posterior regions of the brain showed ‘compression’, mostly due to changes in shape [10].

The stiffness of the brain is low, resulting in the shifting of brain tissue when there are changes in head tilting relative to gravity, even in normal, healthy individuals who are not having surgery [10]. Zappalà et al. [10] found that repositioning healthy young adults from the prone to supine position caused the posterior fossa to be compressed. This movement has the potential to compress brainstem autonomic nuclei, predisposing them to ischemia formation there. Other changes in head positioning may have unexpected effects on brain shifting. Much more research is needed to quantify these and their potential application to heart failure patients.

### 2.4. Spatial Relationships of Brain Components

Another investigative group from King’s College London [11] looked at the accuracy of MRI to document the spatial relationships of different parts of the brain relative to each other and to consistently find those same relationships in a subsequent follow-up MRI performed at a later time if, in fact, they were unchanged. They considered several variables such as the make and model of the MRI machine, time interval between scans, and geographical placement of the patient on the surface of the scanner bed (i.e., slightly toward the starboard, port, bow, or stern) as well as head tilting (pitch) relative to the scanner bed. They concluded that the only variable that caused a small but significant change in the position of brain components relative to each other was head-tilting [11]. This is consistent with gravitational changes in the brain, which the King’s College investigators did not mention.

In a second recent paper [12], the King’s College investigators looked at physiological parameters, such as relative dehydration occurring diurnally, that might affect intra-brain structural relationships. They found none. However, they did not include the time of day differences, which would reflect the sleep cycle, as well as the gradual diurnal development of gravitational ischemia.

### 2.5. Auto-Regulation

Does auto-regulation play a role in this setting? Auto-regulation is a mechanism by which cerebral blood flow is maintained, primarily through regional vasoconstriction and vasodilation within the brain. It does not respond well to physical barriers such as intravascular clots or extravascular mass lesions. Gravity behaves as an extravascular mass lesion because it forces extravascular overlying brain tissue from above to ‘push downward’ against the external walls of the blood vessels. Reciprocally, it also forces the underlying skull and meninges to ‘push back upward’ against blood vessels, causing them to be ‘squeezed’ between two forces. Another physical barrier, an intravascular clot, may similarly challenge the ability of autoregulation to maintain cerebral blood flow. Autoregulation often fails in that setting, resulting in the appropriately named ‘cerebrovascular accident’.

Additionally, autoregulation is mediated largely through the brainstem autonomic nuclei. The MRI findings of the Welsh investigators [10] suggested these may occasionally be compressed and that ischemia there may be the likely result. While ischemic, they may lose their ability to function normally to maintain auto-regulation. Lastly, CO_2_ changes may easily and immediately overcome cerebral autoregulation. A healthy adult who rapidly takes five deep breaths will typically begin to experience light-headedness.

### 2.6. Neuronal Railways in the Brain

The Neuroscience–Pathophysiology researchers [4], when examining the brainstems of heart failure patients who died suddenly, considered the following two levels of neural operation.

Events at the cellular biology level within the solitary tract nucleus regarding hyper-excitation.The larger context (Figure 2) regarding the electrical wiring (neuro-anatomy) of sensory receptors of the vagus nerve sending upstream signals from the myocardium, reporting violations of normal physiological parameters during heart failure. Then, after transiting through the brainstem, how does exiting downstream stimulation get routed either to the heart to initiate an arrhythmia or to the pulmonary vasculature to initiate pulmonary edema? The answer to this latter question is unknown.

Hyper-stimulation and hyper-excitation are phenomena of cell biology. Let us step back for a second.

In major cargo transport hubs across the globe, several train lines converge and slowly run side-by-side with each other for only one mile. Along that short span, towering straddle-lifters have their giant feet planted on both sides of the rail tracks, with their cranes carefully positioned to lift entire cargo loads out of train cars heading for City A before dropping them gently into train cars heading to City B.

Astrocytes are the straddle-lifters of the brain [13]. Neuro-transmitter packets are the cargo that they move. Astrocytes, a type of glial cell, also control the switches and traffic lights to briefly open one train route and close another. However, in the brain, two trains running side-by-side are separated by a narrow, but significant, chasm called the synaptic cleft as if riding along stone embankments on opposite sides of a deep canal.

Electricity runs through the rail tracks spanning the entire length of the neuron. Trains circle around the synaptic border, where they give up their cargo to train cars on the opposite side of the synaptic cleft. The complexity of cargo transfer is multiplied in the brain, as real train routes are relatively 2-dimensional, and real trains are uniform in size, but neuronal pathways are very 3-dimensional and non-uniform.

Hyper-excitation occurs in neurons and groups of neurons when surrounding areas are switched off and affected areas are switched on, in terms of delivery of neuro-transmitters, in quantities that can up-regulate intra-cellular metabolism to a point that exceeds the energy supply. The suddenly over-used metabolic apparatus may be one that is used to normalize blood pressure in the setting of heart failure.

### 2.7. Limitations

It is unknown why specific types of cardiac arrhythmias (atrial vs. ventricular) follow abnormal neural–electrical discharges from the dorsal motor nucleus of the vagus nerve into the cardiac conduction system. These specific arrhythmias probably relate to local cardiac anatomy and pathophysiology. Both atrial and ventricular arrhythmias have been observed to occur following these abnormal discharges, but often not in the setting of heart failure, which is not the only medical condition to elicit them.

The occasional evolution of a cardiac arrhythmia from ‘atrial’ into the generally more lethal ‘ventricular’ type probably occurs due to events and circumstances which are entirely within the heart, as opposed to being influenced by the brain.

Most cardiac arrhythmias generated by abnormal discharges from the dorsal motor nucleus are non-lethal and do not result in death. As such, these are never studied as autopsy cases, and so many details of the neuropathology surrounding their occurrence remain unknown.

## 3. Conclusions

During the 90 days following hospitalization for acute heart failure, 2 mechanisms of sudden unexpected death have been suggested to possibly result from 4 significant influences on pathophysiology in the brain.The two causes of death are severe cardiac arrhythmia [5,7] and neurogenic pulmonary edema [8]. They are both mediated through the brainstem autonomic nuclei generally and executed specifically through the dorsal motor nucleus of the vagus nerve.The four significant influences on pathophysiology in the brainstem autonomic nuclei are:
Hyper-stimulation of neurons in the solitary tract nucleus, increasing their metabolic rate;Inadequate blood flow at a vascular watershed terminus;Additionally decreased blood flow following vasoconstriction related to relative hyperventilation and decreased pCO_2_ levels;Gravitational ischemia in the brain, caused by the weight of brain mass sitting above the brainstem.Changes in head tilt release gravitational ischemia in the brain. There is no specific head position (relative to gravity) that is considered favorable or unfavorable for an extended period of time (i.e., more than 24 h). Even a small degree of head elevation, used in managing pulmonary congestion, may increase gravitational ischemia in the posterior fossa and brainstem.

In this paper, we suggest a new and important research avenue for intervening in cardiac arrhythmias and preventing their occurrence through the non-invasive use of head-tilting and other head repositioning maneuvers using a hospital bed and possibly other equipment. It could potentially help many geriatric patients with heart failure who have decreased mobility in bed and who tend to stay in one position longer, thereby increasing gravitational ischemia in the brain.

No clinical research has yet been performed regarding a brief, transiently maneuvered resolution of gravitational ischemia in the brain for the prevention of life-threatening autonomically-mediated complications in acute heart failure. However, it might seem logical to begin by testing head-tilting maneuvers that have an established track record (both for safety and success) in treating benign positional vertigo (spinning sensation with imbalance) and related conditions, thought by many to involve disruption of fluid flow in the semicircular canals of the inner ear.

Head tilting exercises ‘re-train the brain’ to respond normally to positional changes. Several theories have been suggested as to why these maneuvers are effective, but none have been well substantiated. Gravitational ischemia in the brain may be an etiological component of some vertigo-related conditions, which has not previously been considered.

Anatomically, vertigo emanates from the vestibular nucleus in the medulla, adjacent to the autonomic nuclei. The fact that gravitational forces are so heavily used by the brain specifically in its balance mechanism suggests that other, more subtle uses of gravity may be engaged in nearby neural structures, such as gravitational ischemia in the autonomic nuclei.

In studying head tilting techniques for use in heart failure patients, a five-minute session probably needs to be repeated hourly by nursing or physical therapy staff during the waking hours of the day. Named after the US otolaryngologist John M. Epley, MD, the Epley Maneuver places a pillow underneath the back and between the shoulders. This elevation of the lungs might be appropriate for many heart failure patients.

## Figures and Tables

**Figure 1 diagnostics-12-01473-f001:**
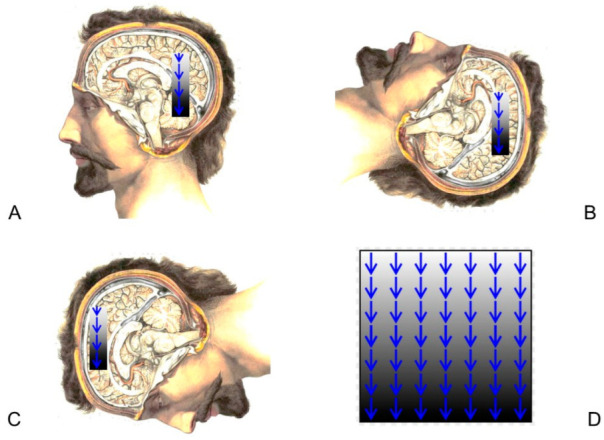
(**A**) Mid-sagittal view of the brain in relation to eye, face, and neck. Vertical (upright) position. Gravity (arrows) is pushing the cerebral hemispheres toward the brainstem and cerebellum. (**B**) Horizontal supine position. (**C**) Horizontal prone position. (**D**) Gravity. Schematic stratification of biological tissue into horizontally pancaking layers under the influence of gravity. Lower layers incur a progressively increasing weight burden from upper layers, thus increasing compression of blood vessels and reduction of blood flow, possibly resulting in regional ischemia. Modified from Wikipedia; creative commons license. https://en.wikipedia.org/wiki/Human_brain#/media/File:Human_brain.jpg accessed on 5 June 2022.

**Figure 2 diagnostics-12-01473-f002:**
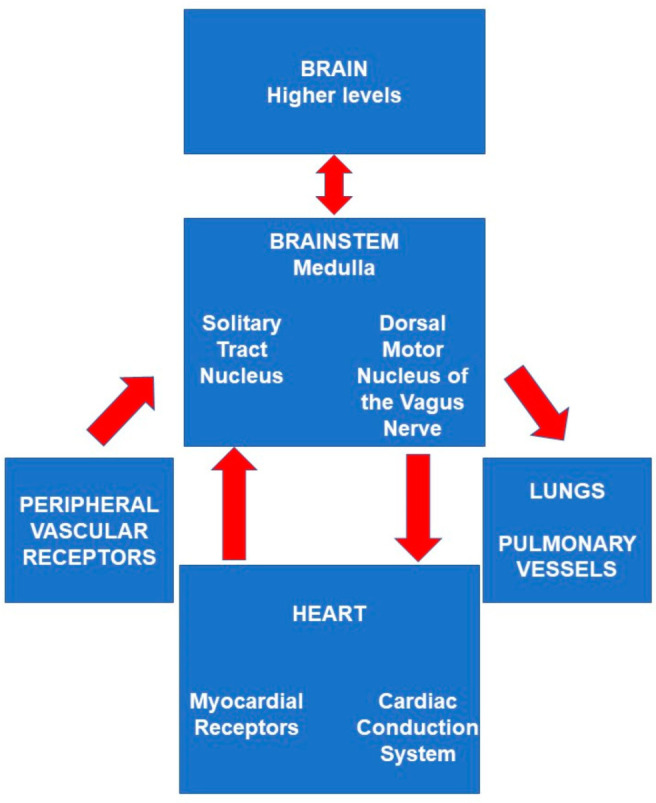
A pathophysiological box diagram shows approximate neuronal circuitry by which abnormal cardiac physiological parameters related to heart failure (i.e., from myocardial stretch receptors) are transmitted to the brainstem. There, they are influenced by higher brain centers before they return to the heart by a neural route which is intended for use by beneficial neural stimulus, but which is occasionally co-opted by a harmful stimulus.

## Data Availability

Not applicable.

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
