# Peer review of "Acute Heart Failure, 90-Day Mortality, and Gravitational Ischemia in the Brain"

_diagnostics, 2022, doi:10.3390/diagnostics12061473_

Round 1

Reviewer 1 Report

The authors approach to explaining Neuronal Railways in the Brain is very interesting.

However, the article of the authors has a number of publications describing gravitational cerebral ischemia in the context of various diseases that lead the patient to hospitalization and bind him to the supine position. It seems that various information is added to the main body of the article about a particular disease. Is it so?

Author Response

We thank the Editor and the Reviewers for their constructive comments.  Replies to individual comments are stated below each comment. The reviewer’s comments are shown in bold font and our responses are shown in italics. Our latest modifications to the manuscript have been highlighted in yellow.

TO REVIEWER # 1

  1. The authors approach to explaining Neuronal Railways in the Brain is very interesting.

Response:  We thank Reviewer 1 for commenting favorably regarding Neural Railways in the Brain.

  1. However, the article of the authors has a number of publications describing gravitational cerebral ischemia in the context of various diseases that lead the patient to hospitalization and bind him to the supine position. It seems that various information is added to the main body of the article about a particular disease. Is it so?

Response: We thank and acknowledge that Reviewer 1 correctly observed several citations of papers describing gravitational ischemia in the brain.  This was a major focus of our paper—as suggested in our title.  

Some of these papers described MRI findings in normal healthy subjects, documenting normal gravitational physics in the brain.  These did not 'report findings in the context of various diseases', but they might have usefulness in a variety of disease states. 

Also, these patients were not bound to the supine position—however, supine position is admittedly a common default position for hospitalized patients, who are often too sick to stand up, and who are often most easily managed in face-up position.

All of the papers cited revolved around the topic stated in our title:  acute heart failure, mortality, and gravity in the brain. 

We hope to have exhaustively responded to all comments. Many thanks for the pertinence and accuracy of the comments.

Sincerely,

 Howard Jaster, MD

Giulia Ottaviani, MD, PhD

Reviewer 2 Report

This review investigates the implication of brain gravitational ischemia in heart failure. This kind of overview is useful in an under known vascular subject in continuous evolution. The review is overall objective, well organized and written. Reading is set to a to a broad audience.

Before publication I have one minor suggestions to make:

- In the conclusion it would be fit to add a section highlighting a treatment for this phenomenon. This would give a cutting edge to the article. Suggestion would also be enough.

Author Response

We thank the Editor and the Reviewers for their constructive comments.  Replies to individual comments are stated below each comment. The reviewer’s comments are shown in bold font and our responses are shown in italics. Our latest modifications to the manuscript have been highlighted in yellow.

TO REVIEWER # 1

  1. The authors approach to explaining Neuronal Railways in the Brain is very interesting.

Response:  We thank Reviewer 1 for commenting favorably regarding Neural Railways in the Brain.

  1. However, the article of the authors has a number of publications describing gravitational cerebral ischemia in the context of various diseases that lead the patient to hospitalization and bind him to the supine position. It seems that various information is added to the main body of the article about a particular disease. Is it so?

Response: We thank and acknowledge that Reviewer 1 correctly observed several citations of papers describing gravitational ischemia in the brain.  This was a major focus of our paper—as suggested in our title.  

Some of these papers described MRI findings in normal healthy subjects, documenting normal gravitational physics in the brain.  These did not 'report findings in the context of various diseases', but they might have usefulness in a variety of disease states. 

Also, these patients were not bound to the supine position—however, supine position is admittedly a common default position for hospitalized patients, who are often too sick to stand up, and who are often most easily managed in face-up position.

All of the papers cited revolved around the topic stated in our title:  acute heart failure, mortality, and gravity in the brain. 

TO REVIEWER # 2

  1. This review investigates the implication of brain gravitational ischemia in heart failure. This kind of overview is useful in an under known vascular subject in continuous evolution. The review is overall objective, well organized and written. Reading is set to a to a broad audience.

Response:  We thank Reviewer 2 for commenting favorably that our paper was well written for a broad audience. 

  1. Before publication I have one minor suggestion to make:

- In the conclusion it would be fit to add a section highlighting a treatment for this phenomenon. This would give a cutting edge to the article. Suggestion would also be enough.

Response: We agree with Reviewer 2 that the Conclusion should be expanded slightly to include mention of specific approaches to treatment.   We have added the following paragraph to our paper:

“No clinical research has yet been performed regarding a brief transiently maneuvered resolution of gravitational ischemia in the brain for the prevention of life-threatening autonomically-mediated complications in acute heart failure.  But it might seem logical to begin by testing head-tilting maneuvers that have an established track record (both for safety and success) in treating benign positional vertigo (spinning sensation with imbalance) and related conditions, thought by many to involve disruption of fluid flow in the semicircular canals of the inner ear.

Head tilting exercises 're-train the brain' to respond normally to positional changes.  Several theories have been suggested as to why these maneuvers are effective, but none have been well substantiated.  Gravitational ischemia in the brain may be an etiological component of some vertigo-related conditions---and this has not previously been considered.

Anatomically, vertigo emanates from the vestibular nucleus in the medulla, adjacent to the autonomic nuclei.  The fact that gravitational forces are so heavily used by the brain specifically in its balance mechanism suggests that other more subtle uses of gravity may be engaged in nearby neural structures, such as gravitational ischemia in the autonomic nuclei. 

In studying head tilting techniques for use in heart failure patients, a 5-minute session probably needs to be repeated hourly, by nursing or physical therapy, during the waking hours of the day.    Eponymous after US otolaryngologist, John M. Epley, MD, the Epley Maneuver places a pillow underneath the back, and between the shoulders.   This elevation of the lungs might be appropriate for many heart failure patients”.

[Conclusions, pages 8-9, lines 340-365]

We hope to have exhaustively responded to all comments. Many thanks for the pertinence and accuracy of the comments.

Sincerely,

 Howard Jaster, MD

Giulia Ottaviani, MD, PhD

Round 2

Reviewer 1 Report

The authors answered all my questions. Thanks.